# Autoencoder Spectral Unmixing for Single-Cell Raman Analysis

Emil Alstrup Jensen[*1], Lasse Ebdrup Pedersen[2], Anders Kristensen[3], and Line Clemmensen[1,4]

[1]DTU Compute, Technical University of Denmark, Kgs. Lyngby, Denmark
[2]DTU Bioengineering, Technical University of Denmark, Kgs. Lyngby, Denmark
[3]DTU Health Tech, Technical University of Denmark, Kgs. Lyngby, Denmark
[4]Department of Mathematical Sciences, University of Copenhagen, Copenhagen, Denmark
ealje@dtu.dk

## Abstract

Raman spectroscopy provides label-free, holistic molecular information at the single-cell level, but spectra are complex and challenging to interpret. We apply physics-constrained autoencoders with volume regularization to unmix single-cell Raman spectra, guiding latent dimensionality and promoting diverse, interpretable endmembers. On THP-1 and NK cell datasets, this approach improved peak definition, chemical interpretability, and captured biologically relevant variability.

## 1 Introduction

Raman spectroscopy (RS) is a label-free and non-invasive analytical method that probes molecular composition and structure by detecting how light interacts with molecules. Recently, RS has attracted attention for single-cell analysis. Unlike many existing approaches, as it provides holistic information spanning genes, proteins, and metabolites [1]. However, each Raman spectrum is a complex superposition of numerous molecular signatures, making interpretation challenging. Deep learning, particularly physics-constrained autoencoders, can extract meaningful information without requiring pure reference spectra [2] and often outperforms classical unmixing methods like MCR [3], VCA [4], and N-FINDR [5]. In autoencoders, latent dimensions correspond to molecular components. While performance can be validated for simple systems with known references, single cells contain tens of thousands of molecules across wide concentration ranges, making reference-based validation and selection of latent dimensionality challenging. In this work, we extend physics-constrained autoencoders for single-cell Raman analysis by incorporating a volume loss that guides latent dimensionality and promotes diverse, interpretable endmembers.

## 2 Methods

**AE-based spectral unmixing:** Let $\mathbf{x} \in \mathbb{R}^M$ be a Raman spectrum measured at $M$ Raman shifts. Each spectrum is assumed to be a mixture of $N$ pure components (endmembers), with $N << M$. The goal of unmixing is to recover the endmember matrix $E = [\mathbf{e}_1, ..., \mathbf{e}_N] \in \mathbb{R}^{M \times N}$ and the abundance vector $a \in \mathbb{R}^N$ for each spectrum. An autoencoder is trained with an encoder $\mathcal{E} : \mathbf{x} \to \mathbf{a}$ and decoder $\mathcal{D} : \mathbf{a} \to \hat{\mathbf{x}}$ such that $\hat{x} \approx x$. Here, we consider a simple linear mixing model:

$$\mathbf{x} = E\mathbf{a} = \sum_{n=1}^{N} a_n \mathbf{e}_n$$

This is physically reasonable with baseline correction [6] and global normalization, preserving relative contributions in the dataset. The autoencoder is trained using RMSE reconstruction loss and to encourage diverse endmembers, a volume loss is added:

$$\mathcal{L} = \mathcal{L}_{\text{rec}} + \mathcal{L}_{\text{vol}} = \|\mathbf{x} - \hat{\mathbf{x}}\|_2^2 + \lambda(\det(EE^{\text{T}} + \epsilon I) + \epsilon)^{-1/2}$$

Where $\lambda$ determines the strength of the volume regularization. Abundances are enforced to be non-negative using a softplus-tanh transformation of the encoder outputs, while endmembers are kept non-negative by clipping during training.

The encoder consists of parallel 1D convolutional layers (kernel sizes 3, 5, 16 filters each), to capture sharp peaks and broader spectral features. Their outputs are concatenated, linearly projected, and passed through a fully connected layer with 128 units, followed by a final layer mapping to the latent bottleneck of dimension $N$.

**Datasets:** Two low SNR datasets are used in this study, characterized by many of the challenges previously described regarding prior knowledge, labeling, and biochemical complexity. A single-cell dataset containing a large set of Raman spectra from primary Natural Killer (NK) cells (acquired by the authors) resolved by approximately 1 pixel per cell ($N_{spectra}$=205900). And a dataset consisting of volumetric RS raster scans of a human leukemia monocytic (THP-1) cells ($N_{spectra}$=64000) [7].

**Evaluation metrics:** We evaluate the autoencoder performance on the THP-1 cell dataset using the following spatial resolution metrics. The pixel-wise abundance variance:

$$\text{spatial\_var} = \frac{1}{HW} \sum_{x=1}^{H} \sum_{y=1}^{W} \frac{1}{N} \sum_{n=1}^{N} \left( a_{x,y,n} - \bar{a}_{x,y} \right)^2$$

---

*Corresponding Author.

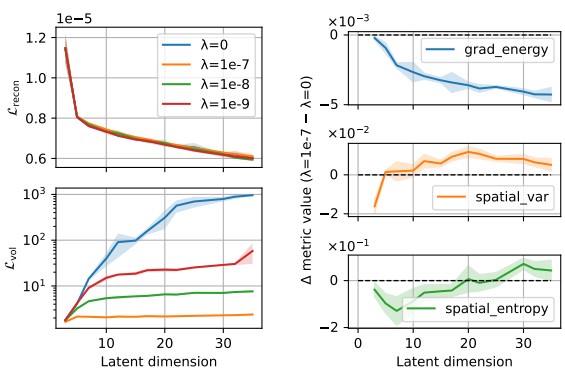

**Figure 1.** Left: Validation reconstruction loss and post hoc computation of volume loss. Right: Difference between resolution metrics with and without volume regularization (across 5 random seeds).

Where H and W are the image dimensions and $\bar{a}_{x,y}$ is the mean abundance at pixel (x,y). The Shannon entropy of all abundance values:

$$\text{spatial\_entropy} = - \sum_{i=1}^{N_{\text{bins}}} p_i \log(p_i)$$

Where $p_i$ is the fraction of abundance values falling into bin $i$. The average squared spatial gradient of each abundance map:

$$\text{grad\_energy} = \frac{1}{N} \sum_{n=1}^{N} \frac{1}{HW} \sum_{x=1}^{H} \sum_{y=1}^{W} \left( \nabla a_{x,y,n} \right)^2$$

## 3  Results

We evaluated autoencoder-based unmixing with ($\lambda$=1e-7) and without ($\lambda$=0) volume regularization. Endmembers were matched using spectral angle distances and the Hungarian algorithm [8]. For both datasets, reconstruction and volume losses were tracked across bottleneck dimensions. Reconstruction loss behaved similarly with and without volume regularization (see Fig. 1 for THP-1 dataset). For the THP-1 dataset, endmembers generally contain peaks that are more well-defined (see Fig. 2). At certain latent dimensions (5 and 20), volume regularization seems to improve spatial variance and entropy in cell images, indicating that more subtle details are resolved (see Fig. 1). The gradient energy seems to always be larger for $\lambda$=0 and decrease at higher latent dimensions, reflecting a smoother, more distributed encoding across endmembers. So volume regularization increases detail at the cost of less spatial contrast for individual endmembers. Similarly, for the NK cell dataset volume regularization seems to promote diverse endmembers with well defined peaks, which makes chemical interpretation easier. As seen in Fig. 3 distinct peaks from carotenoids (1155 and 1510 cm$^{-1}$) and from

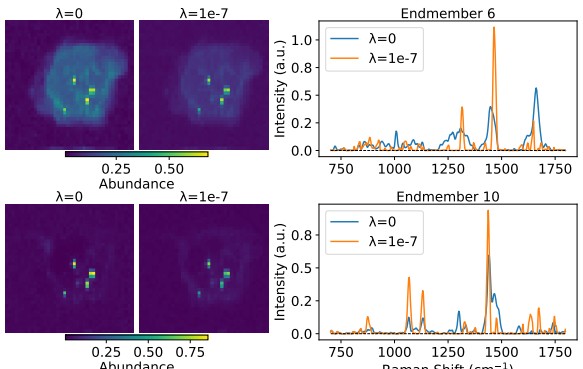

**Figure 2.** THP-1 cell dataset. 2 of the 20 endmembers with corresponding abundance maps.

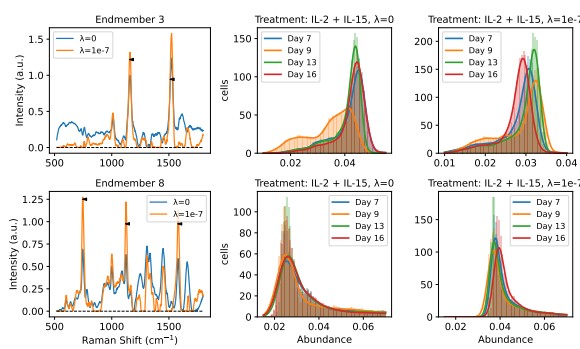

**Figure 3.** NK cell dataset. 2 out of 10 endmembers with corresponding abundance histograms labeled by time point in a cell expansion/activation study for a specific treatment with IL-2 and IL-15 cytokines.

cytochrome c (745, 1120 and 1577 cm$^{-1}$) can be used to characterize endmembers 3 and 8, respectively. Volume regularization enhances the peaks with respect to the rest of the spectrum, and even improves separability between biologically relevant labels (days in expansion/activation) when looking at abundances in each cell.

## 4  Conclusion

Physics-constrained autoencoders with volume regularization enable effective spectral unmixing of complex single-cell Raman data. Volume regularization guides the selection of an appropriate latent dimensionality and promotes diverse, interpretable endmembers. On both THP-1 and NK cell datasets, it improved peak definition and chemical interpretability while capturing biologically relevant variability. These results demonstrate the potential of autoencoder-based approaches for analyzing high-dimensional single-cell Raman spectra. In future work, the volume regularization should be implemented in an adaptive way, so the choice of latent dimension does not depend on post hoc analysis.

## Acknowledgments

This study was supported by the DigitSTEM cooperation/Research agreement with DTU (to LC), funded by Bioneer A/S, non-for-profit research-based organization.

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
