# OpenReview forum: "Autoencoder Spectral Unmixing for Single-Cell Raman Analysis"
_NLDL.org/2026/Abstracts_Track — NLDL 2026 Abstracts_

### Official Review · Reviewer_Abpa · 2025-10-27

**Soundness:** 3
**Correctness:** 4
**Rating:** 5
**Confidence:** 3

**Summary:**

The authors demonstrate strong expertise in coding by applying autoencoders to their model, emphasizing the potential of volume regularization for analyzing single-cell Raman spectra. The abstract aligns well with the conference’s thematic directions.

**Strengths:**

Cell analysis was conducted using an autoencoder-based unmixing approach with (λ=1e-7) and without (λ=0) volume regularization. This strategy enabled a more efficient identification of characteristic peaks in cancer cells after fine-tuning the parameters of the proposed model.

**Weaknesses:**

As illustrated in Fig. 2, further improvement could be achieved by applying additional denoising, histogram equalization, or post-classification to the obtained images for more accurate result estimation.

In the text, variables (x, y), H, and W are italicized.
Abbreviations are expanded as follows: RMSE, MCR, VCA, N-FINDR.

Post-analysis of the data can help physicians perform evaluations more efficiently: the radius of risk occurrence be extracted separately for future screenings, providing specific information about each patient’s individual response to potential recurrence.

---

### Official Review · Reviewer_xBfa · 2025-10-28

**Soundness:** 4
**Correctness:** 4
**Rating:** 5
**Confidence:** 5

**Summary:**

The authors propose to use physics-constrained autoencoders with volume regularization to unmix single-cell Raman spectra.

**Strengths:**

This is a highly relevant and challenging application. Two datasets are used in the study, the method is clearly explained in the limited space given in an abstract submission and results look very promising. The application area is very interesting and insights can be valuable to a broader range of researchers than just this very application.

**Weaknesses:**

The abstract does not mention any release of the code or data, which would be of great value in terms of reproducibility and future research.

---

### Official Review · Reviewer_Qz8P · 2025-10-28

**Soundness:** 4
**Correctness:** 4
**Rating:** 5
**Confidence:** 4

**Summary:**

The abstract presents a proposed improvement in physics-constrained autoencoder based spectral unmixing of Raman spectra in single-cell analysis. By adding a volume loss as a regularization, authors showcase improved peak definition and interpretability on two datasets.

**Strengths:**

Well presented results.

**Weaknesses:**

The structure and shape of the datasets is a little unclear.

---

### Decision · Program_Chairs · 2025-11-05

**Decision:**

Accept

**Comment:**

The abstract is of interest to the community and should be presented at the conference.